# Vibration-Based Non-Contact Activity Classification for Home Cage Monitoring Using a Tuned-Beam IMU Sensing Device

**DOI:** 10.3390/s25082549

**Published:** 2025-04-17

**Authors:** Pieter Try, René H. Tolba, Marion Gebhard

**Affiliations:** 1Department of Electrical Engineering and Applied Sciences, Westphalian University of Applied Sciences, 45897 Gelsenkirchen, Germany; pieter.try@w-hs.de; 2Institute for Laboratory Animal Science and Experimental Surgery, University Hospital, RWTH Aachen University, 52074 Aachen, Germany; rtolba@ukaachen.de

**Keywords:** inertial measurement unit, sensor fusion noise reduction, signal processing, vibration sensing, activity monitoring, activity classification, home cage monitoring, machine learning

## Abstract

This work presents a vibration-based non-contact monitoring method to classify the physical activity of a mouse inside a home cage. A novel tuned-beam sensing device is developed to measure low-amplitude activity-induced cage vibrations. The sensing device uses a mechanical beam structure to enhance a six-axis IMU that increases the signal-to-noise ratio (SNR) by 20 to 40 times in a relevant environment. A sophisticated classification algorithm is developed to process vibration sequences with a variable time frame that utilizes multi-level discrete wavelet transformation (MLDWT) to extract time–frequency features and optimize signal properties. The extracted features are classified by a convolutional neural network–long short-term memory (CNN-LSTM) machine learning model to determine the activity class. The ground truth is obtained with a camera-based system using EthoVision XT from Noldus and a custom post-processor. The method is developed on a dataset containing 300 h of vibration measurements with camera-based reference and includes two separate home cages and two individual mice. The method classifies the activity types Resting, Stationary Activity, Walking, Activity in Feeder, and Drinking with an accuracy of 86.81% and an average F1 score of 0.798 using a 9 s time frame. In long-term monitoring, the proposed method reproduces behavioral patterns such as sleep and acclimatization as accurately as the reference method, enabling home cage monitoring in the husbandry environment with a low-cost sensor.

## 1. Introduction

In recent years, vibration analysis has been used in a wide range of applications, from predictive maintenance, where machine vibration is analyzed to predict failure [1,2], to structural health monitoring, where the condition of buildings and civil structures is assessed [3,4]. Lately, this approach has been used to monitor the activity of various living creatures by analyzing the vibration of ambient objects that are generated by activity. Every activity has a characteristic set of movements that generates vibrations with unique properties and patterns. The unique vibrations enable activity classification and are generated by dynamic forces that are applied to ambient objects as a result of movement. The main benefits of the method are the low cost of the sensing equipment, no requirement for physical contact or line of sight, and lower computational requirements than, e.g., camera-based methods. In previous work, this concept has been demonstrated in various applications such as human occupancy detection [5], foot step localization [6], the detection of various human activities [7,8], the classification of physical activity of pigs [9] and activity monitoring of mice [10]. The primary challenges are the measurement of small-amplitude activity-induced vibrations and the signal processing for activity classification.

The advantages of vibration-based activity monitoring are particularly well suited to enable high-throughput home cage monitoring of mice in the husbandry environment. Large-scale mice husbandry is found in research facilities, which accommodate thousands of animals in individual home cages and do not have a cost-effective method to monitor a large number of cages. The state of the art is to move animals from their home cage to dedicated observation environments to measure behavior and activity using cameras [11]. This is very labor-intensive and results in significant bias as well as additional stress for the animals caused by human handling. There are several methods proposed for the home cage scenario, which use radar [12], cameras [13,14,15], capacitive sensors [16], implanted RFID chips [17,18], infrared barriers [19] and combinations of the mentioned. However, none have found widespread use due to cost, complexity, propriety or lacking capability to distinguish specific activities. In contrast, vibration-based monitoring is capable of classifying specific activities, uses low-cost sensors, and requires less computing power than camera-based methods. A major application is the automatic assessment of animal welfare, which supports the caretakers in their daily workflow by enabling on-demand caretaking and improved scheduling. Furthermore, this method offers a novel tool for research and provides previously unattainable behavioral data in the home cage environment, where bias is considerably reduced. This leads to a refinement in the quality of life and a reduction in the amount of animals needed, serving the three Rs of animal welfare [20].

In previous work [10], the concept of vibration-based activity monitoring in the home cage scenario is initially explored, and a first proof of concept is achieved. It employed a six-axis micro-electromechanical system (MEMS) inertial measurement unit (IMU) mounted to the bottom of the cage floor to measure cage vibration in the three spatial directions with the accelerometers and gyroscopes of the IMU. Employing an MEMS IMU provides key benefits which are its small size, required by the constrained space, low-cost, resulting in high scalability, and high sampling rate to measure vibrations in the kilohertz range. The previous work was able to discern the activity classes resting, stationary activity, and locomotion by analyzing 15 s vibration sequences with a Fourier-based classification algorithm. It used a camera-based reference method that tracked the mouse and determined the activity class based on movement velocity. It is shown that each sensor of the IMU captures unique and useful information that is beneficial for activity classification. The proof of concept was able to accurately differentiate activity from non-activity but was limited in differentiating stationary activity and locomotion. It is found that the amplitude of activity-induced vibrations created by mice was very small. This was compounded by the bedding which covers the ground and dampens force impulses and vibrations. Combined with the comparably high intrinsic noise of IMUs, vibration measurements had a very low signal-to-noise ratio (SNR), which is a major challenge to improve performance.

In this work, we aim to improve on previous methods and achieve robust vibration-based activity monitoring in the home cage environment. The major contributions of this work are the introduction of a tuned-beam vibration sensing device, the development of a robust vibration classification algorithm, and the design of a reliable reference method for activity. This work introduces the tuned-beam sensing device first presented in our previous work [21] to measure activity-induced vibrations with a high SNR. The device consists of an IMU mounted at the tip of an optimized mechanical beam structure, which was able to increase the SNR of vibration measurements by many times in laboratory tests with a calibrated force impulse generator. However, this approach requires precise tuning to the operational environment for best results, which can be challenging due to variable conditions. This work introduces an experimental tuning procedure to overcome this challenge and guarantee optimal performance under any condition. Furthermore, a sophisticated activity classification algorithm is developed that is designed to analyze the unique data of the tuned-beam sensing device. The device outputs six data streams of three accelerometers and three gyroscopes with a sampling rate of 7 kHz, which exhibit resonant characteristics due to the beam construction. The classification algorithm has a multi-level discrete wavelet transformation (MLDWT) preprocessing stage that extracts and augments time–frequency features and a convolutional neural network–long short-term memory (CNN-LSTM) classification network to determine the activity class. These components are complemented by a robust reference method that combines the commercial behavior analysis software EthoVision XT 17.5 from Noldus, seated in The Netherlands [14], with a custom post-processing algorithm to generate reliable activity labels based on top-down video recordings. The proposed method is developed and evaluated with measurements obtained in the central animal facility of the University Hospital RWTH Aachen. The measurements include vibration measurements with video reference data of two home cages, with each housing one mouse over a time period of seven days for a total of about 300 h of data. The results verify that the tuned-beam sensing device increases the SNR in the relevant environment by about 40 times, enabling the extraction of activity-related information. In combination with the reliable reference method and sophisticated classification algorithm, a robust and accurate vibration-based activity classification is achieved, and long-term monitoring is demonstrated with a low-cost sensor.

## 2. Materials and Methods

In this work, we aim to achieve robust high accuracy vibration-based activity monitoring in the home cage environment, which has the major challenges of measuring the low-amplitude activity-induced vibrations and classifying these measurements to determine the activity class. Several components are required for the proposed method, which will be presented in the following chapters. First, the setup is presented that is used to obtain the dataset of home cage vibrations and video-based reference data. The setup consists of two standard home cages, which are equipped with the tuned-beam sensing device. The employed sensing devices were designed and experimentally tuned with a novel tuning procedure for optimal performance. Afterwards, the reference method is presented that analyzes videos of an infrared (IR) camera mounted in the lid of the cages to obtain a ground truth of the activity. The reference method uses the behavior analysis software EthoVision and a custom post-processing algorithm to analyze the videos. Five activity classes are defined based on the capabilities of the reference method. Following this, the activity classification algorithm is presented, which is organized into a preprocessing step and a classification step. The preprocessing step uses MLDWT to extract time–frequency features and enhance signal quality by fusing multiple sensor axes. The classification step employs a highly optimized CNN-LSTM network to analyze the complex local and long-term patterns contained in activity-induced vibrations.

### 2.1. Experimental Setup for Data Acquisition

An experimental setup is designed to measure activity-induced vibrations of a home cage with video-based reference of the true activity. For this study, two standard home cages (Zoonlab HRC500 [22]) were equipped with sensors and monitored for a week each. The home cages are shown in Figure 1. Each home cage is 40.8 cm long, 18.3 cm wide, 11.6 cm tall, and has an internal area of about $500\;{\mathrm{cm}}^{2}$ without the lid. The main components are the vibration sensing device mounted on the bottom and an IR camera mounted in the lid. The sensing device is based on the commercial ASM330LHHTR six-axis IMU [23]. In accordance with the guidelines, the home cages were additionally equipped with a food rack, a water drinking bottle, filtered air vents, and a cotton nestlet for nest building. The home cages were kept in an observation room that was climate-controlled and had automated lighting to simulate a circadian rhythm.

As part of this study, activity-induced cage vibrations were measured in a standard animal room in the central animal facility of the University Hospital RWTH Aachen. This was an observational study only; there was no intervention with the living space inside the home cage and no pain, suffering, or distress. Therefore, the study was below threshold and defined as no animal experiment; thus, approval was not required. This statement and study were also reviewed by the Internal Animal Care and Use Committee (IACUC) of the University Hospital RWTH Aachen and in line with the conception of the Governmental Body, the Landesamt für Umwelt-, Natur- und Verbraucherschutz (LANUV) NRW, which shared the assessment given here.

In this study, two home cages were used to evaluate the impact of mechanical variations in different home cages. Each cage also contained a different mouse to evaluate the impact of individual behavioral differences of mice. Furthermore, the length of the monitoring period was maximized to evaluate the natural variation within distinct activities. The individual home cages, their tuned-beam sensing devices and the mice within were numbered for future reference. Mouse 1 was a male CD56/Black6 weighing approximately 33 g. Mouse 2 was also a male Black6 weighing around 36 g. The male mice were separated before because of aggressive behavior.

#### Vibration Sensing Device

The tuned-beam vibration sensing device is an essential component and is employed to measure the low-amplitude activity-induced vibrations with a high SNR. It uses a highly optimized mechanical beam structure, which is integrated in the printed circuit board (PCB) of the sensing device to amplify the vibrations measured by the IMU. The IMU sits at the tip of the beam, which is mounted to the cage with a spacer. A disadvantage of this approach is that unexpected mechanical variations in the operating environment can affect the performance of the device, for example, when a new batch of home cages is slightly different in weight or size. This work introduces an experimental tuning procedure to reduce the impact of these variations to optimize performance. The tuning procedure is used to adjust the vibration characteristics of the beam structure in the final operational environment for maximum performance. The vibration properties of the beam structure are adjusted using a tuning weight attached to the tip of the beam structure just below the IMU sensor. The first step is to determine the theoretically ideal geometry of the beam using finite element method (FEM) simulation. The ideal geometry is shortened by a millimeter to ensure that the manufactured beam has a resonant frequency that is too high and that can be adjusted by adding weight, which is much easier than removing weight. When the sensing device is manufactured and mounted on the cage, the tuning weight is optimized iteratively by adding and removing small temporary weights based on vibration amplification measurements. The vibration amplification is determined with a calibrated force impulse generator, described in [21], that applies a predetermined series of force impulses to the cage, during which the signal energy is measured and evaluated. This procedure determines the required tuning weight to an accuracy of ±0.025 g. Finally, the multiple temporary weights are replaced with a single steel weight which is glued in place. The mass of the large weight is finely adjusted by varying the amount of the thermoplastic glue.

Figure 2 shows a tuned-beam sensing device with an installed tuning weight. The geometry of the tuned beam has a length of 28 mm measured at the inner side of the two lateral bridges which each have a width of 3 mm. Sensing device 1 has a tuning weight of 0.55 g and device 2 has a weight of 0.45 g. The performance of the tuned-beam sensing devices is shown in Section 3.1.

### 2.2. Video-Based Reference Method

The experimental setup employs an IR camera to capture reference videos of the mouse’s activity inside the home cage. The videos are analyzed afterwards to obtain activity labels with timestamps, which is done by tracking the body and analyzing its movement. Physical activity is highly complex and requires extensive knowledge regarding the animal’s behavior to correctly identify different activities. In this work, the commercial and widely used behavior analysis software EthoVision from Noldus [14] is employed in conjunction with a custom post-processing algorithm, presented in Section EthoVision Post-Processing and Activity Class Definition, to achieve a reliable ground truth. Video-based methods are the gold standard for activity analysis but generally require a specific environment for optimal performance. EthoVision is primarily designed for observation environments, which are optimized for camera analysis. Typically, observation cages are opaque rectangular boxes with an open top and high walls, which allows the camera to be mounted high above the cage to avoid wide-angle lenses. Observation cages are also devoid of food pellets, water bottles, or bedding because they are not required during the short stay. In contrast, home cages require bedding in the form of wooden chips that cover the floor, a feeder, food pellets, and a drinking bottle that make tracking more challenging. Home cages are also transparent, which causes undesirable reflections. In addition, the home cage requires a lid due to the low wall height, which limits the vertical height of the camera and requires a fish-eye lens that introduces significant camera distortion. These conditions are very challenging for the camera-based behavior analysis software, but EthoVision offers several options to achieve the best results regardless. The settings described in the following paragraphs are selected based on experimental optimization.

EthoVision labels activity based on the pose, movement, and location of the mouse. It tracks several body parts and uses a map of important areas inside the cage to determine the context of each movement. The map is called arena settings and shown in Figure 3, where important areas such as the floor, walls, eating area, and drinking area are highlighted. A contour-based body point detection is chosen to estimate the center point, nose point and tail–base point of the mouse. The contour is determined using dynamic subtraction of the background with an automatically updated background image. The software outputs activity labels and activity-related variables with timestamps and a sampling rate of 30 Hz, which corresponds to the video frame rate. The most important behavioral variables are shown in Table 1.

#### EthoVision Post-Processing and Activity Class Definition

EthoVision analyzes and outputs activity labels with a rate of 30 Hz corresponding to the camera frame rate. To use this reference data with our proposed vibration-based method that analyzes vibration sequences with a duration of 3 to 11 s, a post-processing algorithm is devised to synchronize the two methods. This is achieved by analyzing the activity labels from EthoVision that correspond to the vibration sequence of a data point to obtain a single activity label that summarizes the activity in that section. The activity summary represents the activity that is most prevalent in a section. If a section contains similar amounts of multiple activities, it is labeled as Uncertain due to its ambiguity and later excluded from evaluation to ensure the quality of the dataset.

The post-processing algorithm uses a decision tree structure, shown in Figure 4, and analyzes the relative duration of each activity class. The relative duration is calculated by dividing the duration of each activity class by the duration of the analyzed section. The decision tree uses thresholds to determine the activity class that is most prevalent in a section. A threshold of 67% is chosen initially, therefore requiring a 2/3 majority in a section for an activity to be chosen as the output label. This threshold is chosen based on the high variability of activity, which results in most sections containing at least a few different activities. This phenomenon is shown in Section 3.2. Increasing the threshold would reduce ambiguity because sections would contain higher amounts of a single activity class. But this also reduces the dataset size, because more data points would be labeled as Uncertain. Reducing the threshold increases the dataset size but creates more low-quality data points with higher ambiguity.

It is not feasible to optimize the thresholds statistically in this work, because it would require a large amount of additional ground truth data that would be obtained by hand annotation. Obtaining a statistically representative amount of hand-annotated data is extremely labor-intensive and outside the scope of this work. Instead, about 80 min of hand-annotated labels were obtained, which are not statistically representative, and manually analyzed to identify common error cases and devise solutions that were implemented in the decision tree. This hand-annotated ground truth was also used to identify activity classes that were unreliable and not suitable for further evaluation.

The decision tree shown in Figure 4 defines five activity classes: Resting, Stationary Activity, Activity in Feeder, Walking, Drinking, and Uncertain. Data labeled as Uncertain are ambiguous and exempt from further evaluation. The activity Drinking was highly reliable and detected when the nose point was inside the water area. It used a threshold of 67%. The activities grooming, sniffing, and eating exhibited a high error rate based on the hand-annotated ground truth data. This is in part due to the similar visual appearance of these activities and the suboptimal camera placement. However, this group of activities was well distinguished from other activity classes, and they were grouped together in the decision tree to define a new activity class called Stationary Activity. It was determined by adding the relative duration of all three classes before applying a threshold of 67%. The activity Walking occurred very often and was often combined with other activities. To reduce its ambiguity, the threshold was increased to 80% for this activity class. The activity Resting was characterized by the absence of movement. However, very low intensity Stationary Activity was often misclassified as Resting. This was counteracted by implementing an additional condition, which required the average velocity during the section to be lower than 0.05 cm/s. In addition, the threshold was increased to 80%, because Resting generally occurs over longer periods of time.

Furthermore, it was found that the mice liked to climb the feed rack and stay inside it for various activities. It is hypothesized that the feed rack heavily influences the vibrations generated by activity. Therefore, a dedicated class was defined to identify these cases called Activity In Feeder. Any activity involving movement that interacted with the feeder was labeled as Activity In Feeder. A 20% threshold was used based on the location label FeedArea_CP (Center point is in area of the feeder). Resting inside the feeder was still classified as Resting.

### 2.3. Activity Classification Algorithm

The vibration measured by the tuned-beam sensing device is a direct result of physical activity and is generated by dynamic forces which are applied to the cage by movement. Key to achieving activity classification is to detect vibration characteristics and patterns that are specific to certain activities, whereby pattern refers to the timing, repetition, and relative amplitude of vibrations. The basic concept of the algorithm is to analyze a sequence of measurements from the IMU by extracting time–frequency information and then interpreting the extracted features using a machine learning method. Each IMU sequence consists of six data streams, which are the three acceleration measurements $a_{x}$, $a_{y}$, and $a_{z}$ and the three gyroscope measurements $\omega_{x}$, $\omega_{y}$, and $\omega_{z}$. Analyzing the dataset of activity-induced vibrations, it was found that the signals $a_{x}$, $a_{y}$, $a_{z}$, and $\omega_{y}$ contained activity-induced vibrations with sufficient SNRs and were used for activity classification. The other sensor signals were omitted for now.

The structure of the proposed classification algorithm is illustrated in Figure 5 and consists of a preprocessing step, where activity-related information is extracted from the data sequences, and a CNN-LSTM classification network that generates the final result. During the preprocessing step, selected sensor signals of the IMU are fused to reduce the sensor noise, as described in [21]. This sensor fusion algorithm is able fuse signals that are correlated by the oscillation mechanics of the beam structure. For example, the primary oscillation at around 100 Hz where the beam bends vertically causes the acceleration in the *z* axis and the rotation around the *x* axis to be correlated.

#### 2.3.1. Preprocessing of Vibration Data

The main task of the preprocessing step is to extract activity-related information and prepare it for ML-based classification. The relevant features are the frequency, amplitude, timing, and pattern of activity-induced vibrations. These are extracted with MLDWT, which is a multi-resolution wavelet analysis technique able to extract frequency as well as time information. MLDWT is used to deconstruct a signal into multiple levels of detail and approximation components, each containing sequential signal components of a specific scale, i.e., frequency. Multi-resolution refers to the fact that each level is derived by further decomposition of only the approximation sub-band at each subsequent level. This leads to a graduated scale, where each detail component has a different frequency range size.

This work decomposes signals to the 10th level, which corresponds to a frequency of 6.8 Hz, using a Coiflet wavelet with five vanishing points and a Tukey (tapered cosine) window function. The remaining approximation coefficient is omitted, which contains the signal components below 6.8 Hz. The decomposition results are collected in a cell array data structure where each element contains the array of a detail component of a specific level and sensor. It is noted that, at this stage, each detail coefficient has a different array length due to the multi-resolution analysis.

Following the decomposition, all detail components are resampled to a rate of 200 Hz with a max pooling operation: (1)$$n=\left\{1,\ldots ,N\right\}\;m=\left\{1,\ldots ,M\right\}\;M=200\;\mathrm{Hz}T_{timeframe}$$(2)$$\mathrm{for}\;M\;<\;N:\;x^{\prime }\left[m\right]=max\left(x\left[\left\lfloor \left(m-1\right)\frac{N}{M}+1\right\rfloor :\left\lfloor m\frac{N}{M}\right\rfloor \right]\right)$$(3)$$\mathrm{for}\;M\;\geq \;N:\;x^{\prime }\left[\left\lfloor \left(n-1\right)\frac{M}{N}+1\right\rfloor :\left\lfloor n\frac{M}{N}\right\rfloor \right]=x\left[n\right]$$
where x[n] is the original detail component with 1 to N elements, $x^{\prime }\left[m\right]$ is the resampled detail component with 1 to M elements, and $T_{timeframe}$ is the vibration analysis time frame. This step equalizes the vector length of the detail components and allows the components to be arranged in a matrix, which is required by the input layer of the CNN-LSTM network. In addition, this step removes redundant information and sensor noise in high frequency detail components. The sampling rate of 200 Hz was derived from the dataset and is sufficient to capture the fastest rate at which vibration is generated by activity, which was observed to be about 10 Hz. Finally, the detail components are normalized, which improves the performance of neural networks. The resulting matrix represents one data point in the dataset.

The sensor fusion algorithm, described in full detail in [21], is embedded in the activity classification algorithm to reduce the sensor noise of applicable detail components. The algorithm structure is illustrated in Figure 5, where it is placed adjacent to the primary wavelet deconstruction. The fusion algorithm operates by extracting certain detail components, fusing them together, and then replacing the original signals with an enhanced fused version for further processing. The basic working principle is based on the correlation of motion in different axes, which results from certain oscillation modes of the beam structure. For example, at about 100 Hz, the beam oscillates vertically by bending at the base, which causes the end of the beam to move in the *z* axis and rotate around the *x* axis. This motion creates a nearly identical signal in the *z*-axis accelerometer and the *x*-axis gyroscope, which have the same frequency but with a different amplitude and phase. The sensor noise contained in the measurements is uncorrelated between sensor axes and normally distributed, because it mainly originates from the thermal noise of the microstructures. By scaling and shifting the signals of the different axes to align the correlated signal components, the signals can be fused by weighted averaging to reduce the uncorrelated sensor noise. This method requires prior knowledge, which is provided by the fusion array. It contains a set of information for each group of correlated signals, which are the frequency of the oscillation mode, the relative phase between the axes, and the amplitude transformation coefficients. The first step is to retrieve the detail components of each correlated sensor axes at the specified frequency. Detail components of sensor axes without a phase difference are retrieved from the primary wavelet deconstruction. Detail components of sensor axes with a phase difference are first phase shifted in the time domain and then deconstructed separately as follows: (4)$$MLDWT\left(x^{\prime }\left[n\right]\right)=MLDWT\left(x\left[n-\left\lfloor \frac{\phi F_{S}}{2\pi f_{osc}}\right\rfloor \right]\right)$$
where ϕ is the phase difference, $f_{osc}$ is the frequency of the specific oscillation mode, and $F_{S}$ is the sampling rate of the IMU. Afterwards, the detail components are scaled in amplitude by multiplying the detail component of each sensor axis with its corresponding transformation coefficient contained in the fusion array. The transformed detail components are then combined by the Adaptive Combiner, which calculates the average of all detail components weighted by their noise variance. The result is a single-detail component which replaces the original version and has a noise variance of(5)$$c=\frac{\sigma_{1}^{-2}}{\sigma_{1}^{-2}+\sigma_{2}^{-2}},$$(6)$$\sigma =\sqrt{c^{2}\sigma_{1}^{2}+{\left(1-c\right)}^{2}\sigma_{2}^{2}},$$
where σ is the noise variance of the result, and $\sigma_{1/2}$ is the noise variance of the original signals. The ideal case is when $\sigma_{1}=\sigma_{2}$, which results in a noise reduction of $\sigma /\sigma_{1}=0.707$.

As described earlier, the phase of the signals is adjusted in the time domain. This is done to prevent an issue in the previous implementation [21], where signals are differentiated in the time domain to achieve a 90 degree phase shift. This operation applies a high-pass filter which significantly dampens signal components with a frequency of under 500 Hz. As a result, the SNR of the original signals is significantly reduced.

#### 2.3.2. Classification Network

The preprocessing stage presented in the prior chapter extracts the time–frequency features from the sensor signals and formats them in a matrix, where each row represents a sequential signal component of a specific scale and of a specific accelerometer or gyroscope. Activity-induced vibrations exhibit both local and long-term patterns. A CNN-LSTM network is designed to extract these activity-related patterns and generate an activity classification. It uses a 1D convolutional neural network (CNN) layer to extract local patterns, the output of which serves as input for a long short-term memory (LSTM) network that extracts long-term patterns in the data and makes the final prediction. The hyperparameters are optimized iteratively based on total accuracy. The final network structure is as follows:Sequence Input Layer;Convolution 1D Layer (Filter Size = 100, Number of Filters = 100);Bidirectional LSTM Layer (Hidden Units = 500, Output = Sequence);Dropout Layer (Probability = 0.2);Bidirectional LSTM Layer (Hidden Units = 500, Output = Last);Fully Connected Layer;Softmax Layer.

The networks were trained using the Adam optimizer with a maximum of 100 epochs. The initial learning rate was set to 0.002, and a mini-batch size of 50 was used. A gradient threshold of 1 was applied to manage gradient clipping. A training/testing split of 70/30 is used to validate the results.

## 3. Results

The dataset for the study was recorded within a one-week period during which two home cages with one mouse in each were monitored simultaneously for a total of 293.3 h/12.2 d of data. The measurements consist of six data streams from the IMU, containing accelerometer and gyroscope data, and video recordings with synchronized timestamps. The videos were analyzed with EthoVision, as described in Section 2.2, which outputs the behavioral analysis with a sampling rate of 30 Hz. Individual data points are generated by dividing the continuous vibration data and behavior analysis output with a uniform analysis time frame without overlapping. For each data point, a true activity label is determined with the EthoVision post-processing algorithm described in Section 2.2, and the vibration data are processed into a feature matrix as described in Section 2.3. The method is designed for a variable time frame to adjust the sampling rate of the method and evaluate the impact of the time frame length.

Table 2 illustrates the support of the activity classes with a time frame of 9 s. The support varies depending on the time frame, but the relative support remains mostly consistent. The support of each activity class differs significantly and is dependent on the individual behavior. The mouse in Cage 2 rested 7% more than the mouse in Cage 1 and walked half as often. It drank less often and spent more time in the feed rack. A total of 31.4% of the data points in Cage 1 were labeled as Uncertain, meaning they could not be assigned to any activity class, which is about 12% higher than in Cage 2. All in all, about 25.6% of labels were Uncertain and excluded in further training and testing.

It is evident that the dataset is imbalanced, with Resting occurring most often. As Resting is generally characterized by a lack of movement, the resulting vibrations are expected to have a low statistical variation. In order to reduce redundant data and training time, the majority class Resting was randomly undersampled. Random data points with the label Resting were excluded until the support of Resting was equal to that of Stationary Activity, which was the second most common activity class. This should not negatively impact the class based on its low statistical variation.

### 3.1. Performance of the Sensing Device

The tuned-beam vibration sensing device was evaluated in a laboratory environment with a highly accurate force impulse generator [21]. It was used to apply a sweeping series of force impulses to home cages that were outfitted with the proposed sensing device. During the series of force impulses, the vibration was measured, and the signal energy was evaluated for each individual force impulse. Based on the signal energy, different sensing devices were evaluated, including the two manufactured tuned-beam sensing devices and a directly mounted IMU. The signal energy for each force impulse was determined by analyzing the accelerometer measurements over a period of 0.2 s—beginning at the moment when the force impulse was applied. It is calculated as(7)$$E=\sum_{t=0\;s}^{0.2\;s}{\left(\sqrt{a_{x}^{2}+a_{y}^{2}+a_{z}^{2}}\right)}^{2},$$
where *E* is the signal energy, and *a* the acceleration in different directions.

Figure 6 illustrates the results of the evaluation, showing the energy of vibration of the two tuned-beam sensing devices in orange and a directly mounted IMU in blue. The plotted values are the average results for 20 measurements. By dividing the signal energy of the tuned-beam devices by the signal energy of the directly mounted sensor, the amplification of vibration was obtained. It is illustrated by black diamonds and gray circles on the right *y* axis. The signal energy graphs exhibit a flat section on the left which corresponds to the energy of the sensor noise. As the force impulse increased at about 200 µNs, the signal energy graphs rose steadily, while the graphs of the tuned-beam devices rose more steeply. The graphs of the tuned-beam devices have a similar course, with a slight deviation in the region of 500 µNs to 1250 µNs. The amplification graphs illustrate a rising trend with a pronounced peak at around 800 µNs, reaching an amplification of about 350% to 400%. The maximums of the two peaks are slightly offset.

These results verify the efficacy of the proposed tuned-beam sensing device in a laboratory setting and demonstrate a significant enhancement of sensitivity to vibrations caused by force impulses, which effectively increases the SNR.

#### Noise Reduction of the Sensor Fusion Algorithm

The oscillation modes of the tuning-beam sensing device suitable for sensor data fusion are analyzed below. Suitable oscillation modes generate correlated signals in multiple sensor axes with similar SNRs. Equation (Equation 6) shows that the SNR has to be similar to yield a useful noise reduction. Two suitable oscillation modes were found with a frequency of 94 Hz and 906 Hz, where $a_{z}$ and $w_{x}$ were correlated as well as $a_{y}$ and $w_{x}$. The fusion arrays of the two oscillation modes are(8)O(1)={[0,0,1,0.039688,0,0],[0,0,0,π/2,0,0],94Hz},(9)O(2)={[0,1,0,0.027865,0,0],[0,0,0,π/2,0,0],906Hz},
where the first array is the transformation coefficients, and the second array is the phase difference. In the available dataset, an average noise reduction of about 6% for the first mode and about 0.8% in the second mode were achieved. This is within the margin of error of the previous work.

### 3.2. Analysis of Activity-Induced Vibrations

At first glance, the recorded vibration data reveals a highly complex relationship between activity and vibration. Activity-induced vibrations exhibit significant variation in amplitude, frequency composition, and pattern, even within a single activity class. As a result, it is unfeasible to model the correlation between vibration and activity with physical modeling methods. However, drinking and eating are exceptions and produce characteristic vibration patterns.

Figure 7 illustrates vibration measurements and activity labels over a period of 2.5 min to illustrate several activities, including Drinking. EthoVision detected Drinking very reliably based on the nose point and the static position of the drinking nozzle, which were shown in Figure 3. During drinking, a distinctive vibration pattern emerges which is characterized by a series of high-amplitude oscillations that rapidly decay and occur at a frequency of approximately 10 Hz. In Cage 1, this vibration pattern was very consistent and almost always occurred during drinking. In Cage 2, this distinct vibration was less consistent and missing or less pronounced more often.

EthoVision was not able to detect eating reliably due to its similar appearance to other stationary activities. However, a few instances of eating were manually evaluated, where distinct vibrations were observed. These were similar to the vibrations occurring during Drinking but appeared to have a higher variation in amplitude.

The vibration measurements in the dataset were further evaluated to determine the average signal power for each activity class. As vibration is directly correlated to movement, and a certain degree of correlation between signal power and activity class is expected. This is investigated by calculating the signal power of the acceleration magnitude for each data point and averaging these results based on the activity class. The signal power is calculated from the magnitude of the acceleration in the three axes as follows: (10)$$P_{Signal}=\frac{1}{N}\sum_{1}^{N}{\left|\sqrt{a_{x}^{2}+a_{y}^{2}+a_{z}^{2}}\right|}^{2},$$
where *N* is the number of acceleration samples of a data point, and *a* is the acceleration. Table 3 lists the average signal power for each activity class. The unit g represents acceleration based on the earth’s acceleration of $g=9.81\;m/s^{2}$. The average signal power ranged from $8.129\times 10^{-6}\;g^{2}$ to $213.37\times 10^{-6}\;g^{2}$, increasing from the activity class Resting to Drinking. This is because the activity classes are arranged in order of movement intensity. The mean results suggest a correlation of signal power and activity class. However, this impression is undermined by the high standard deviation, which shows that there are large intersections between the classes.

The vibration measurements were further investigated by determining the SNR, which was derived by the ratio of signal power and noise power. The noise power was obtained by evaluating measurements with an empty cage and was $5.075\times 10^{-6}\;g^{2}$. The SNRs listed in Table 3 range from 1.602 during Resting to 42.042 during Drinking. A general SNR was determined by calculating the average of all classes with significant movement, which excluded the Resting class. The average SNR during activity is 19.147, which provides an estimate for the expected SNR of activity-induced vibrations.

### 3.3. Classification Results

In the experimental study, a dataset with continuous measurements was obtained, which was divided into individual data points using a uniform analysis time frame. As a result, the total number of data points depends on the chosen time frame. Five different versions of the dataset were created with different time frames. Each dataset was split 70/30 to create a training dataset and a test dataset which contains unseen data for testing. Afterwards, five networks were trained and tested on their respective dataset.

Table 4 illustrates the test results of the five datasets listing the total support, accuracy, macro average F1 score, and weighted average F1 score. The macro average F1 score is the average F1 score of all classes, and the weighted average is weighted by support of each class. The results achieved with the different time frames exhibit a small difference, varying at most by 3.5% in accuracy and 0.097 in the macro average F1 score. The dataset using a 9 s time frame achieved the best results, with an accuracy of 86.81% and a macro average F1 score 0.798. The lowest result was observed by the dataset using a 3 s time frame, with a macro average F1 score of 0.701.

Table 5 illustrates the class-specific results of the dataset using the 9 s time frame. The results show a significant difference in the performance of activity classes. Resting achieved the highest F1 score of 0.903 and is followed by Stationary Activity and Drinking, which achieved F1 scores of 0.8681 and 0.8475. The lowest F1 score was observed in Walking, with a score of 0.6132. The datasets using other time frames exhibit similar class-specific test results.

The confusion matrix shown in Figure 8 provides additional insight into the results of the 9 s time frame dataset and illustrates the distribution of false positive and false negative data points. The matrix shows that misclassified data points exhibit a specific distribution. For example, data points with the true label Resting were almost exclusively misclassified as Stationary Activity, ehile data points with the true label Walking, Act. in Feeder, and Drinking were most commonly misclassified as Stationary Activity.

#### Comparison of Datasets of Individual Animals

Two individual mice are represented in the acquired dataset and were monitored for one week each. Analyzing the number of occurrences of activities in Table 2, it is evident that the individuals exhibited different behaviors. A qualitative analysis was conducted by analyzing videos, vibration data, and EthoVision activity labels, which indicates the following differences: The mouse in Cage 1 appears to have been more active overall. It walked more, drank more, and rested less. Drinking consistently generated strong fast-paced vibrations, as described in Section 3.2. Eating was performed in multiple ways—either by removing a food pellet and eating somewhere on the ground, by eating directly from the feeder while standing on the floor, or by eating while inside the food rack. On the other hand, the mouse in Cage 2 appears to have been less active. It walked less but performed more Stationary Activity overall. During Drinking, the characteristic vibration did not appear as consistently as in Cage 1. This mouse regularly ate inside the food rack and was not observed to remove a pellet to eat elsewhere.

The influence of individual animal behavior on the proposed method was examined by testing its performance separately for each mouse. This was achieved by dividing the dataset into two subsets that contain measurements from a single mouse each. The results of the subsets are shown in Table 6, which lists the performance achieved with each mouse separately. The proposed method achieved higher results on the dataset of mouse 1 with a minimum accuracy of 86.67% and a maximum of 88.58% using a 5 s time frame. On the dataset of mouse 2, a minimum accuracy of 80% was achieved with a maximum of 84.19% using a time frame of 9 s.

The class-specific results of the datasets were compared using the 7 s time frame, which is the time frame at which both datasets demonstrated near optimal performance. The class-specific results of mouse 1 and mouse 2 are shown in Table 7, which lists the accuracy, macro average F1 score, and weighted average F1 score for each class. In the dataset of Cage 1, the classes Resting, Stat. Act., and Drinking achieved the highest F1 scores of around 0.9. The remaining classes scored above 0.52. In Cage 2, the highest F1 scores were observed for Resting, Stationary Activity, and Activity in Feeder. Drinking attained a score of 0.61 and Walking a score of 0.4.

### 3.4. Demonstration of Long-Term Monitoring

The proposed method enables long-term activity monitoring by analyzing consecutive vibration sequences. This capability is demonstrated below using a 9 s analysis time frame, which achieved the best performance when tested on the full dataset. The network for the 9 s time frame was trained on 70% of the dataset.

On a short period of time, individual data points can be visualized directly. Figure 9 presents a 30-min excerpt of activity from Cage 1, comparing the reference method with the proposed method. The activity is depicted in three graphs: (a) and (b) represent the reference method, and (c) illustrates the proposed vibration-based method. The first 10 min of the excerpt are primarily characterized by Resting. This is followed by a period of Stationary Activity, which is interspersed with numerous short instances of Walking and some longer instances of Activity in Feeder and Drinking.

In order to visualize activity over a long period of time, the individual activity classifications were grouped into three-hour intervals and displayed according to their relative occurrence during the interval. The result is shown in Figure 10, which illustrates activity over a period of seven days as a series of stacked bar graphs that illustrate the relative occurrence of each activity class. Data points that were undefined or classified as “Uncertain” by the reference method were excluded. The proportion of different activities is a clear indicator for trends and patterns of activity. Low-activity periods were characterized by a high proportion of Resting, while high activity periods were characterized by a high proportion of Walking, Stationary Activity, Drinking, and Activity in Feeder. The most distinct behavioral pattern is the circadian rhythm, which is clearly observed by the fluctuating levels of Resting. The highest activity was observed around 17:00 to 08:00, with a higher proportion of activities such as Walking, Drinking, and Activity in Feeder. The highest activity level was observed in the first few hours, which gradually decreased over the first two days until it reached a consistent level. This burst of activity represents an acclimatization phase and was attributed to the mice exploring their freshly cleaned home cage to which they were moved to at the start of the monitoring period. The results of mouse 1 and mouse 2 exhibit similar behavioral patterns. Both mice exhibited the acclimatization phase and the circadian rhythm with very similar timing. A major difference was the higher proportion of Activity in Feeder and lower proportion of Walking of mouse 2. This difference is also observed in Table 2.

In comparison to the reference method, the proposed method is able to accurately monitor activity and capture behavioral patterns. It successfully reproduces the circadian rhythm, the acclimatization phase, and arbitrary short-term activity changes with excellent agreement with the reference method. However, the exact number of activity occurrences shows some deviation which is due to classification errors presented in detail in Section 3.3.

## 4. Discussion

The results show that the proposed vibration-based activity classification method is able to distinguish five different activity classes with a total accuracy of 87% and an F1 score of 0.8. It is further shown that the proposed method achieves comparable results in long-term monitoring compared to the camera-based reference method, offering a novel non-contact, high throughput method for activity monitoring. In the following chapter, the test results are discussed to further highlight the general performance and areas of improvement.

### 4.1. Influence of Time Frame Length

The results in Section 3.3 indicate a small but significant difference in the performance when employing varying analysis time frames. There are two major effects that are hypothesized to contribute to these observations.

Activity and the vibrations generated by it vary in both intensity and duration. Consequently, activity-specific vibrations occur intermittently rather than continuously, which can be seen in Figure 7. As the analysis time frame approaches the duration of the breaks in the intermittent vibrations, the likelihood increases that a data point will fall within a break and not contain any activity-induced vibration. As a result, the analysis time frame has a lower limit, which is determined by the activity-specific continuity of vibrations.

The maximum analysis time frame is largely determined by the typical duration of activities. Optimal results are achieved when data points have low ambiguity and only contain one vibration from one type of activity. However, if the analysis time window exceeds the typical duration of activities, the data points will inevitably contain vibrations of different activities.

A potential solution for the time frame limitations is a multi-resolution analysis approach. In this approach, vibration sequences would be analyzed multiple times using the optimal analysis time frame for each activity class. This could improve performance by optimizing the analysis time frame but would significantly increase computational demand.

### 4.2. Error Analysis and Limitations

In Section 3.3, the performance of the proposed method was evaluated in relation to numerous parameters. All in all, the proposed method achieved high performance, but some activity classes underperformed in certain conditions. In the following paragraphs, the most common errors of each activity class are discussed, and methods for improvement are presented.

#### 4.2.1. Class Resting

Overall, the Resting class achieved a high accuracy of 86% and was robustly differentiated from the classes Walking, Activity in Feeder, and Drinking. However, about 14% of the data points were false negatives and mistakenly labeled as Stationary Activity. A major source of this error is likely due to limitations of the reference method. Specifically, EthoVision does not analyze the position of the mice’s extremities. This limitation leads to systematic oversight of subtle body movements during Stationary Activity, which leads to incorrect Resting labels. Additionally, though rare, EthoVision can lose track of the mouse and instead track stationary objects in the background. The presented errors can only be mitigated by improving the reference method. Furthermore, these findings highlight a significant advantage of our proposed IMU-based approach over the vision-based activity monitoring method.

#### 4.2.2. Class Stationary Activity

In the Stationary Activity class, about 5% of the data points were misclassified as Resting. This misclassification likely arises from two main sources. First, some data points that had the true label Resting were incorrectly labeled as Stationary Activity by the video reference method. This was caused by a tracking artifact that introduced jitter to the position of body points, which led to incorrect detection of motion. The second error source lies in the limited sensitivity of the proposed method to detect very small movements which occur more frequently in Stationary Activity. These subtle movements are visually detected but generate minimal vibrations. A notable example is sniffing, where the mouse remains stationary while moving its nose to explore its surroundings. The slow and minor movements produce small dynamic forces that may be insufficient to create detectable structural vibrations. It remains uncertain whether these movements generate sufficient dynamic force to induce vibrations in the cage. However, increasing the sensitivity of the vibration sensor could help mitigate this error.

With a 14.57% result, the false positive rate of Stationary Activity is significantly higher than that of the other classes, which indicates that a large portion of data points were mistaken for this class. Multiple conditions are identified that contributed to this outcome. First, Stationary Activity includes many different behaviors such as sniffing, eating, and grooming, which are grouped into one class because the reference method is unable to distinguish them individually. As a result, a wide range of different vibration patterns are associated with Stationary Activity, which increases its ambiguity. Secondly, some activity types like grooming and eating are contained in two classes, which are Activity in Feeder and Stationary Activity, and are only distinguished by the location of occurrence. The results show that activity in the feeder produces distinct vibrations which requires a dedicated class. But it is very likely that the vibration patterns contain some similarities that contribute to misclassification.

Overall, the Stationary Activity class serves to group multiple types of activity into one class in order to provide a reliable reference, which results in a higher false positive rate. It is recommended to replace the class Stationary Activity with distinct classes for the individual activities sniffing, grooming, and eating.

#### 4.2.3. Class Walking

The activity class Walking performed lower than average, with a F1 score of 0.61. The activity-specific properties were a major contributor to this underperformance. Mainly, the average duration of Walking was much shorter than that of the other classes. It was about a second or less, while the other activities typically lasted for several seconds. Additionally, Walking was almost always accompanied by other activities such as sniffing, eating, and grooming, which occurred between instances of Walking, as seen in Figure 9. As a result, most data points labeled as Walking also contained vibration sequences that were associated with the class Stationary Activity. This is supported by the fact that data points of the class Walking were almost exclusively misclassified as Stationary Activity (44%). This outcome indicates that the vibration analysis time frame is too long for the class Walking. However, it is not feasible to lower the analysis time frame, as it has an adverse effect on the overall performance, as shown in Section 3.3. A possible solution, first presented in Section 4.1, is to implement a multi-resolution approach. This involves analyzing vibration sequences multiple times with different vibration analysis time frames to provide the optimal timeframe for each activity class.

Another reason for the low performance of the class Walking is related to the detection algorithm of EthoVision, which is solely based on the center point of the mouse and its velocity. This approach ignores the extremities and potentially misclassifies non-walking movements as Walking despite the lack of footsteps. It is anticipated that footsteps generate the characteristic vibrations in this activity, which is why this limitation is likely to have a negative impact on the accuracy of this class.

#### 4.2.4. Class Drinking

In Section 3.2, it is observed that Drinking produces a very distinctive vibration pattern, which contributed to its high accuracy. While this vibration pattern occurred reliably when mouse 1 was drinking, it was less reliable for the mouse in Cage 2. Subsequently, the accuracy of Drinking was lower in Cage 2. This is most probably due to a difference in behavior, because mouse 2 did generate this distinct vibration on occasion.

#### 4.2.5. Impact of the Reference Method

In summary, it is concluded that the limitations of the reference method have a major impact on the performance of the proposed method. The main limitations are the lack of full-body pose estimation and the ambiguity of the Stationary Activity class. The results suggest that our proposed IMU-based method can achieve much better performance when combined with a more sophisticated reference method. This includes higher accuracy and the ability to detect additional activity classes that were found to have promising vibrational characteristics. However, improving the reference method requires a significant amount of development and has many challenges to overcome. The prevailing gold standard is manual annotation, a process that requires an insurmountable amount of work and can exhibit significant inter-rater variability. While there are some commercial multi-camera systems that achieve better performance than our reference method, access to them is limited due to high cost and propriety. In order to develop a high-performance video-based reference method from the ground up, open-source tools such as the body-tracking software DeepLabCut [24] could be combined with a dedicated ML-based classification algorithm. However, the main challenge remains the need for manually annotated data for training.

### 4.3. Generalization Performance

This work presents a wide range of results that help to evaluate the generalized performance of the proposed method. In our use case scenario, two conditions are identified that are likely to vary in a real-world setting. These are the physical environment and the mice’s individual behavior, which will be discussed below.

#### 4.3.1. Environmental Factors

The environment in animal husbandry is strictly regulated by guidelines. Home cages are stored in racks which are located in isolated rooms with climate control, special lighting to simulate the day/night cycle, and minimal sources of disturbance. As a result, the environmental conditions are very consistent. The only component that can vary and directly affect the proposed method is the home cage itself, which is not strictly standardized or regulated. However in practice, the variety of home cages is very small. This is because, for practical reasons, all home cages use the same design featuring a deep rectangular plastic tub with a removable lid and side rails for placement in a rack. In addition, most home cages have very similar dimensions, which are derived from the minimum space required per animal. Finally, there are few companies that manufacture home cages. Nevertheless, home cages of different manufacturers have small design variations which affect the vibration characteristics of the cage. In the case of small design differences, the tuned-beam sensing device can be tuned experimentally to achieve optimal results, as proposed in Vibration Sensing Device in Section 2.1. In the case of larger design variations where experimental tuning is not viable, the beam design can be adapted by changing the length and width of the lateral beams. As a general rule, the larger the cage, the lower its resonant frequency, and the longer the beam has to be.

Nevertheless, in Section 3.1, it is observed that the two sensing devices exhibited a small deviation in vibration characteristics, even though they were mounted on home cages of the same model and tuned for optimal performance. This is caused by mechanical variations in the sensing device and cages, which are a result of manufacturing tolerances. Based on the results, it is concluded that the small manufacturing-related discrepancies have no effect on the proposed method. While a small difference in accuracy was observed between Cage 1 and Cage 2 in Section 4.3.2, this difference can be fully attributed to the significant differences in behavior of the two mice.

Considering that there are different home cage models, the adaptability of the classification algorithm must also be considered. The proposed method uses the cage as a transducer to convert the forces caused by the mice’s movement into structural vibrations of the home cage. The amplitude and frequency of the vibration are influenced by the mechanical properties of the cage. However, the cage does not directly influence activity-related characteristics such as the timing and pattern of the vibrations, which contain most of the activity-related information. Therefore, it is hypothesized that a classification model will perform well, even when trained on data of another cage with a similar design. It is beyond the scope of this work, but will be addressed in future work, to train models with data from different types of home cages.

#### 4.3.2. Behavioral Variation of Activity

The previous chapters have discussed the ambiguity of physical activity which arbitrarily varies in duration, intensity, and other metrics. This variability can be classified into two categories which will be discussed in the following.

The first category of variation originates from the arbitrary nature of physical activity. Each activity can be performed in a variety of ways, and its execution is largely arbitrary, although it is influenced by external factors such as the time of day or human presence. The impact of this variation is determined by evaluating data over a long period of time, which is sufficient to capture the full spectrum of each activity. The proposed method achieves high results in these circumstances, demonstrating its robustness against the inherent variation of physical activity.

The second category of variation is a result of the individual behavior, and its effects are documented in Section 3.2 and in the Comparison of Individual Animals in Section 3.3. The unique behavior exhibited by individual mice significantly impacts the manner in which specific activities are conducted. This disparity also influences the proposed method and results in a difference in performance when evaluating the data of each mouse independently from another. However, when the entire dataset was evaluated, incorporating data from both mice, the proposed method demonstrated superior performance compared to its performance on the individual datasets. This outcome suggests that unique individual behaviors can be accommodated if the dataset contains sufficient information about them. Nevertheless, given the limited sample size of individual animals, it is challenging to draw a generalized conclusion regarding the impact of individual behavior. Furthermore, both subjects were from the same strain of mice, which tend to behave similarly.

### 4.4. Comparison to Previous Works

In this work, the tuned-beam sensing device, first proposed in [21], was refined by introducing an experimental tuning procedure and verified in a relevant environment. The results in Section 3.1 verify in a laboratory environment that the proposed experimental tuning procedure effectively increases the performance of the tuned-beam sensing device by nearly double compared to a sensing device without the experimental tuning procedure. Furthermore, the sensing device was successfully employed in a relevant environment to measure the activity-induced vibrations created by mice. Over a period of fourteen days, vibration measurements achieved an SNR upwards of 42.04 with an average of 19.15 during periods of activity.

In the 2022 study [10], vibration-based activity monitoring has been investigated with the same type of home cage using an unmodified IMU directly mounted to the cage floor. The dataset of the previous study consists of about 24 h of data and was analyzed to obtain the activity-specific SNR to derive values for comparison. The values were obtained by calculating the signal power of each data point and averaging based on activity class, as described in Section 3.2.

Table 8 shows that an SNR of 1.019 was achieved in the class resting, 1.059 in stationary activity, and 1.118 in locomotion. It is noted that the activity classes of this previous work were based on a different reference method and are not directly comparable to the ones defined in this work. Nevertheless, it is evident that the tuned-beam sensing device achieved an SNR that is a magnitude higher in every activity class. Comparing the SNR of the classes with the highest values, a 37.6-fold improvement is observed.

Comparing the classification results to the aforementioned work from 2022, this work uses a more sophisticated reference method, which is able to discern more activity classes with a higher accuracy than before. Combined with the high-quality vibration data provided by the tuned-beam sensing device and the wavelet-based classification algorithm, this work presents a method with a much higher accuracy while being able to distinguish more activity types than before. It is evident that all improvements culminate in a method with greater capability and higher accuracy, which has been verified with multiple animals over a longer period.

## 5. Conclusions

In this work, an innovative method is presented to achieve vibration-based activity classification for non-contact activity monitoring of mice in a home cage scenario. A key innovation is a novel vibration sensing device that achieves a high SNR when measuring the low-amplitude vibrations that are generated by the activity of mice. The sensing device employs a tuned mechanical beam structure to enhance a multi-axial MEMS IMU, and its performance has been verified in a relevant environment. Based on this sensing device, a classification algorithm was developed that uses MLDWT to extract activity-related information from the multiple data streams of the IMU and a CNN-LSTM classification network to classify the activity. The algorithm was developed and evaluated on a dataset obtained at the University Hospital RWTH Aachen. The dataset has a total duration of two weeks and contains vibration measurements with the tuned-beam sensing device and video reference of an IR camera mounted in the lid of the cage. Two individual mice, each housed in a separate home cage, are represented in the dataset with a duration of one week per individual. Five different activity classes were defined based on the dataset and specifications of the reference method, which are Resting, Stationary Activity, Walking, Activity in Feeder, and Drinking.

The measurements show that the tuned-beam sensing device reliably captured activity-induced vibrations and achieved a high SNR with an average of 19.15, which is 20 to 40 times higher than that achieved by an unmodified directly mounted sensor. The classification algorithm, tested on unseen data encompassing 30% of the whole dataset, achieved an accuracy of 86.81%, with a macro average F1 score of 0.798, using a 9 s vibration analysis time frame. Among the activity classes, Drinking, Stationary Activity, and Resting demonstrated the highest performance, with F1 scores ranging from 0.847 to 0.903. In contrast, Walking achieved an F1 score of 0.613, while Activity in Feeder scored 0.756. The method is robust against individual behavioral differences and the significant variation observed within each activity class. Furthermore, the proposed method performances comparable to the video-based reference method in long-term activity monitoring and accurately reproduces behaviors such as the sleep cycle, acclimatization phase, and short-term activities.

Overall, the proposed method surpasses its initial goals and achieves robust non-contact activity monitoring comparable in performance to a video-based system. These capabilities have significant implications for mice husbandry and enable high-throughput activity monitoring in home cages, which leads to several improvements such as automatic health assessment and demand-oriented care. It also enables large-scale acquisition of behavioral data in the husbandry sector, which is especially valuable for research purposes and previously unattainable due to high cost and propriety.

Furthermore, this work optimizes and verifies the feasibility of the tuned-beam sensing device in a practical application. This sensing concept presents a powerful and cost-effective tool to enhance IMUs and expand their capabilities in other applications such as seismic measurements or predictive maintenance.

## 6. Future Work

A particular focus was placed on the video-based reference method to obtain a reliable ground truth. For this reason, a commercial behavior analysis software was employed which provides extensive animal knowledge. Ultimately, the reference method was able to discern five activity classes with sufficient reliability, but it did not perform optimally in the home cage environment. As a result, a significant number of observed errors were systematically related to the limitations of the reference method. In recognition of these findings, future research will focus on improving the reference method, which will lead to better performance and expand the capability to detect additional activity classes that exhibit very unique vibration patterns as well.

The results indicate that the individual behavior of mice can affect the performance of the method. Therefore, a broad study is advised with a large number of both genders, including mice of other strains, to offer deeper insight into the range of behaviors and the requirements to achieve a general classification network. This information is crucial to ensure that the method remains robust and precise in real-world application, where facilities house mice of various strains.

Furthermore, given that the proposed method is not limited to mice, it is highly valuable to explore the effectiveness of the method in scenarios involving other living creatures. Future work could extend to species such as rats, rabbits, or humans in other environments, including larger cages, indoor rooms, and even aquariums.

Another promising concept for future research is location estimation. The activity classes Stationary Activity and Activity in Feeder contain identical activities such as grooming and eating, which are distinguished solely by their location. The method’s ability to differentiate between these classes suggests its potential for location estimation. This is based on the unique vibration characteristics that result from interaction with the metal feeder. Additionally, variations in vibration characteristics across different cage locations could further enable precise activity localization.

## Figures and Tables

**Figure 1 sensors-25-02549-f001:**
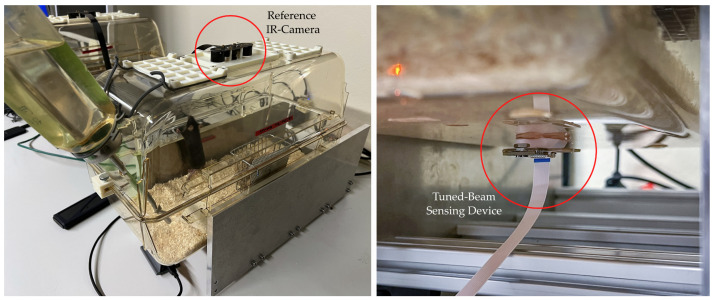
This figure depicts the experimental setup consisting of two Zoonlab HRC500 home cages. The IR camera is mounted in the lid, which is built from an additional HRC500 cage that has been turned upside down. The tuned-beam sensing device is mounted in the center of the cage floor and connected to a micro-controller.

**Figure 2 sensors-25-02549-f002:**
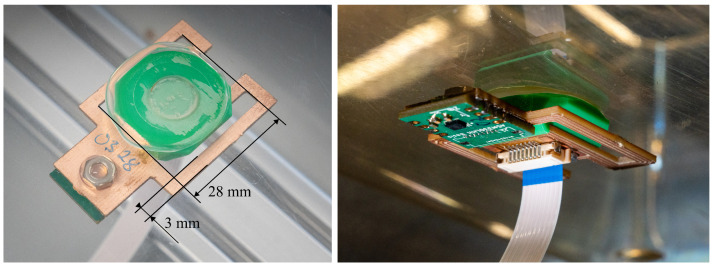
This figure shows one of the tuned-beam vibration sensing devices mounted to the floor of a cage. The left image shows the device from the top where the tuning weight can be seen. The right image shows it from the bottom where the small rectangular IMU component can be seen.

**Figure 3 sensors-25-02549-f003:**
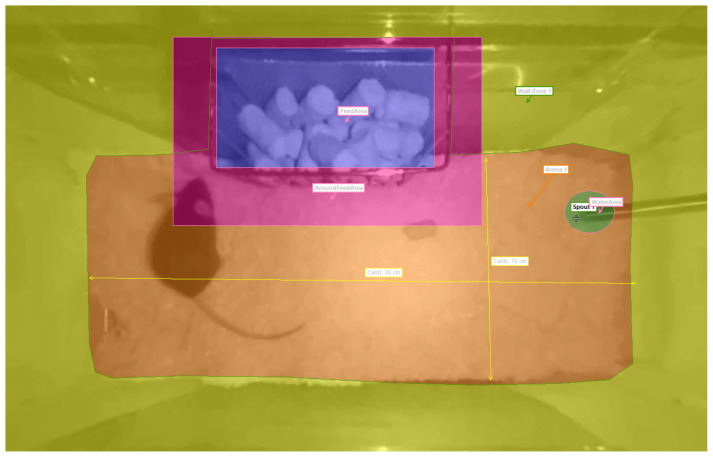
Arena settings used in Noldus EthoVision to detect different activities. The yellow area depicts the cage walls, the orange area the cage floor, the magenta area the vicinity of the food rack, and the blue area depicts the inside of the food rack. The green circle marks the drinking sprout.

**Figure 4 sensors-25-02549-f004:**
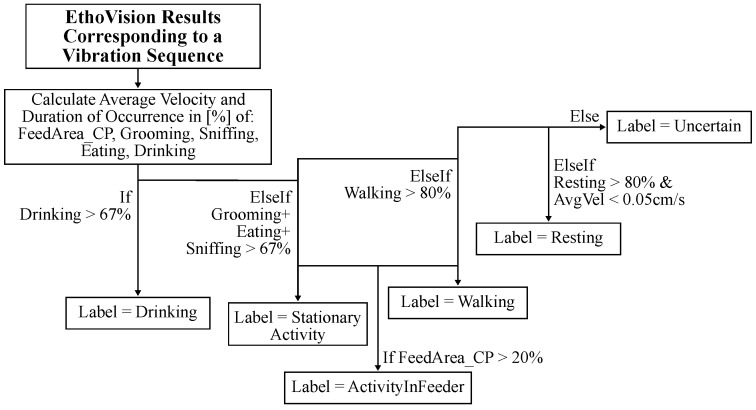
This figure illustrates the decision tree structure of the post-processing algorithm that processes the raw output of EthoVision to obtain a single activity label for a given time frame.

**Figure 5 sensors-25-02549-f005:**
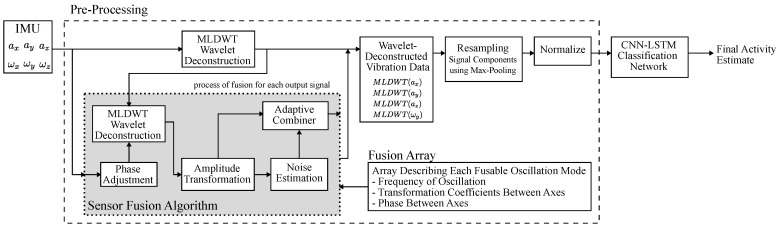
This figure illustrates the structure of the activity classification algorithm.

**Figure 6 sensors-25-02549-f006:**
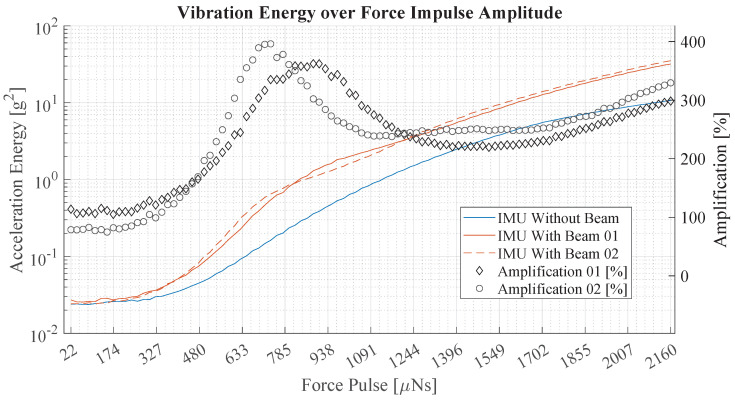
This graph shows the sensitivity of two tuned-beam sensing devices and a directly mounted IMU by plotting the measured signal energy of vibrations created by a force impulse generator. The *x* axis depicts the force impulse amplitude, and the *y* axis shows the signal energy of the acceleration magnitude. The right *y* axis depicts the amplification, which is the ratio of signal energy with the tuned-beam sensing device to the energy with the directly mounted IMU.

**Figure 7 sensors-25-02549-f007:**
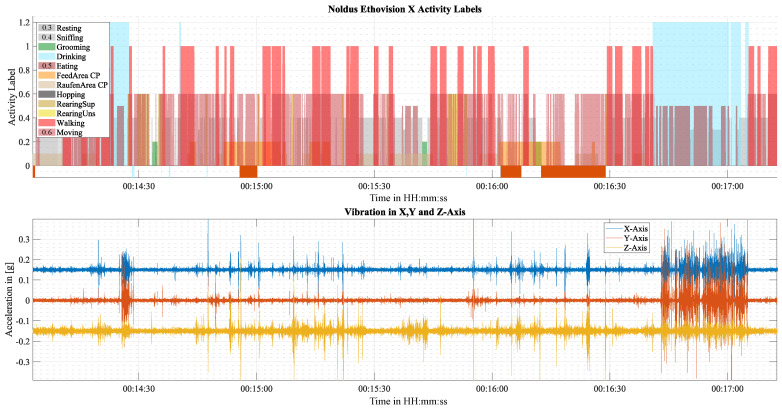
Vibration data of the tuned-beam sensing device and activity labels obtained with EthoVision XT over a time period of about 2.5 min. The upper graph illustrates the unedited analysis of EthoVision XT with different colors and amplitudes for visual clarity. Negative values depict invalid labels caused by, e.g., loss of tracking. The lower graph depicts the acceleration in the *x*, *y*, and *z*-axis with a 0.15 g offset for illustration purposes.

**Figure 8 sensors-25-02549-f008:**
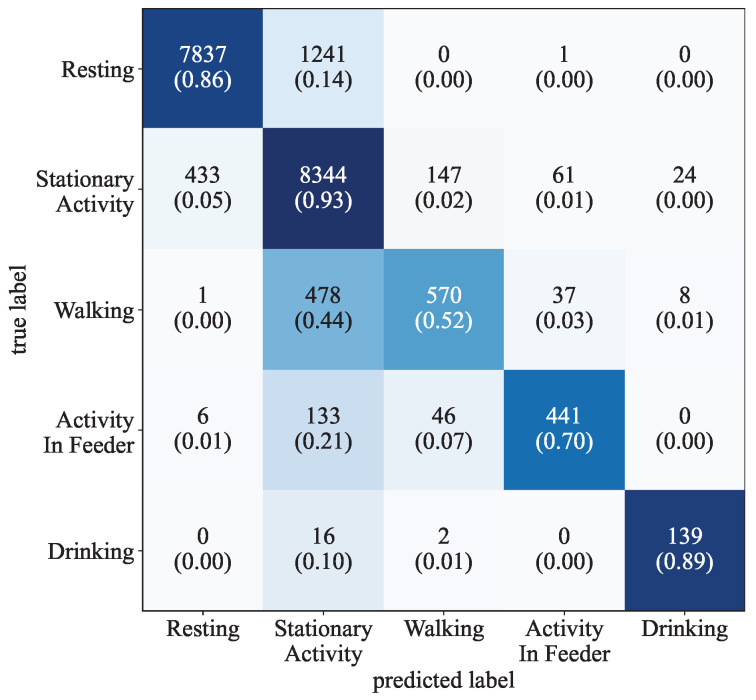
Confusion matrix showing the test results of the 9 s time frame dataset. The upper number in each box represents the total amount of data points. The numbers in parentheses show the relative amount of true positive labels in relation to the total number of true labels, which is also emphasized by the darkness of the blue tone.

**Figure 9 sensors-25-02549-f009:**
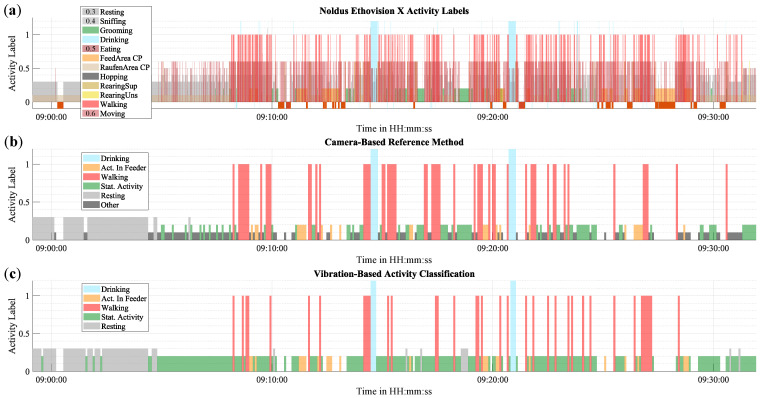
Activity monitoring results of the proposed method in comparison to the reference method over a time period of 30 minutes. Activity classes are illustrated using different colors and amplitudes for visual clarity. Negative values depict invalid data points that are caused by, e.g., loss of tracking. These appear as gaps in graph (**b**,**c**). (**a**) depicts unedited results of EthoVision XT, (**b**) shows post-processed reference labels, (**c**) shows results of vibration-based method.

**Figure 10 sensors-25-02549-f010:**
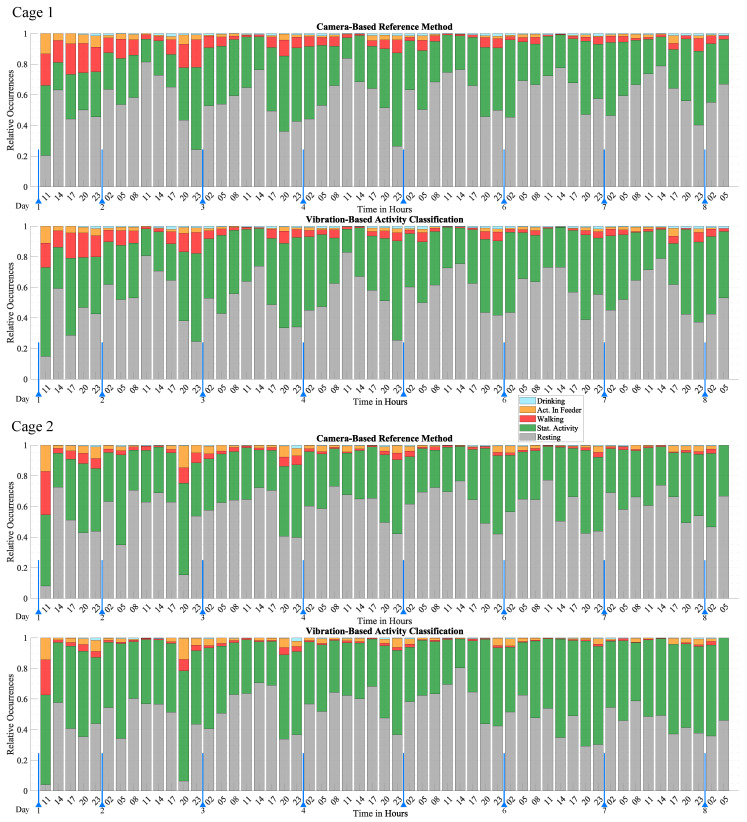
This figure depicts the long-term monitoring capabilities of the proposed vibration-based activity classification method over the available dataset of seven days. The activity is visualized as stacked bar graphs that depict the relative occurrence of activity in three-hour intervals. The blue arrows mark midnight.

**Table 1 sensors-25-02549-t001:** This table illustrates a selection of output variables for activity classification obtained from Noldus EthoVision XT.

Name	Type	Description/Working Principle
Velocity	Variable	Velocity of the center point.
AroundFeedArea_CP	Boolean	Center point is in the area around the feed rack (magenta area).
FeedArea_CP	Boolean	Center point is inside the feed rack (blue area).
Movement	Activity Class	Derived by center point and nose point velocity. Thresholds at 1.5 and 1.3 cm/s.
Drinking	Activity Class	When nose point is in water area.
Grooming	Activity Class	Derived by, i.a., body posture, nose point movement, etc.
Hopping	Activity Class	Hopping. *
Resting	Activity Class	Resting. *
Eating	Activity Class	Eating. *
Sniffing	Activity Class	Sniffing. *
Walking	Activity Class	Derived by, i.a., velocity, location, etc.
Rearing Supported	Activity Class	Derived by, i.a., body stretching, wall area, etc.
Rearing Unsupported	Activity Class	Derived by, i.a., body stretching, etc.

* The detection method for these activities is not disclosed.

**Table 2 sensors-25-02549-t002:** This table illustrates the support of each activity class in the dataset containing measurements from Cage 1 and Cage 2. The data points illustrated here use a time frame of 9 s.

	Resting	Stationary Activity	Walking	Activity in Feeder	Drinking	Uncertain	Total
Cage 1							
Support	23,270	13,258	2323	762	334	18,318	58,265
Relative Support [%]	39.9	22.8	4.0	1.3	0.6	31.4	
Time [h]	58.2	33.1	5.8	1.9	0.8	45.8	145.7
Cage 2							
Support	27,715	16,844	1266	1269	189	11,757	59,040
Relative Support [%]	46.9	28.5	2.1	2.1	0.3	19.9	
Time [h]	69.3	42.1	3.2	3.2	0.5	29.4	147.6
Complete Dataset							
Support	50,985	30,102	3589	2031	523	30,075	117,305
Relative Support [%]	43.4	25.7	3.1	1.7	0.4	25.6	
Time [h]	127.5	75.3	9.0	5.1	1.3	75.2	293.3

**Table 3 sensors-25-02549-t003:** This table lists the average signal power of the acceleration magnitude for each activity class. The table shows results of the whole dataset obtained from data points using a time frame of 9 s.

	Signal Power		SNR	
	Average [10^−6^g^2^]	SD [10^−6^g^2^]	Avg/Noise	[dB]
Sensor Noise	5.075	-	-	-
Resting	8.129	2.66	1.602	2.05
Stat. Activity	21.858	101.91	4.307	6.34
Walking	66.577	104.37	13.118	11.18
Act. In Feeder	91.967	134.44	18.121	12.58
Drinking	213.37	175.34	42.042	16.24

**Table 4 sensors-25-02549-t004:** This table illustrates the test results achieved on the whole dataset. It lists the results of five networks that were trained using different time frame lengths.

Time Frame [s]	Total Support	Accuracy [%]	Macro Average F1 Score	Weighted Average F1 Score
3	209,333	83.31	0.701	0.830
5	122,073	83.03	0.735	0.825
7	86,084	86.79	0.769	0.866
9	66,547	86.81	0.798	0.867
11	54,017	86.09	0.722	0.859

**Table 5 sensors-25-02549-t005:** This table illustrates the class-specific results of the dataset using a 9 s time frame and shows the results of each activity class.

	Precision	Recall	F1 Score	Total Support	Rel. Support
Resting	94.7%	86.3%	0.903	30,202	45.38%
Stat. Act.	81.7%	92.6%	0.868	30,202	45.38%
Walking	74.5%	52.1%	0.613	3589	5.39%
ActInFeeder	81.7%	70.4%	0.756	2031	3.05%
Drinking	81.3%	88.5%	0.847	523	0.79%
Macro Avg.			0.797		

**Table 6 sensors-25-02549-t006:** This table illustrates the average test results of the proposed method for each mouse and time frame separately.

	Cage 1			Cage 2		
**Time Frame**	**Accuracy**	**Macro Avg. F1 Score**	**Weight. Avg. F1 Score**	**Accuracy**	**Macro Avg. F1 Score**	**Weight. Avg. F1 Score**
3 s	86.67%	0.725	0.864	80.00%	0.592	0.802
5 s	88.58%	0.770	0.882	82.05%	0.603	0.814
7 s	87.18%	0.761	0.869	83.64%	0.709	0.833
9 s	88.20%	0.752	0.880	84.19%	0.653	0.837
11 s	88.37%	0.751	0.883	82.05%	0.577	0.816

**Table 7 sensors-25-02549-t007:** Comparison of the class-specific test results achieved with each mouse separately using a time frame of 7 s.

	Cage 1			Cage 2		
**Class**	**Prec./Recall**	**F1 Score**	**Rel. Support**	**Prec./Recall**	**F1 Score**	**Rel. Support**
Resting	96.4%/89.1%	0.92	44.6%	87.9%/85.0%	0.86	46.5%
Stat. Act.	82.7%/93.5%	0.88	44.6%	80.9%/85.8%	0.83	46.5%
Walking	75.4%/46.7%	0.58	7.2%	60.4%/30.3%	0.40	3.0%
Act. In Feeder	50.0%/53.4%	0.52	2.4%	82.2%/84.0%	0.83	3.4%
Drinking	90.3%/91.6%	0.91	1.2%	53.4%/71.6%	0.61	0.6%

**Table 8 sensors-25-02549-t008:** This table illustrates the average signal power and activity-specific SNR achieved in this previous study [10].

	Signal Power		SNR	
	Average [10^−4^g^2^]	SD [10^−4^g^2^]	Avg/Noise	[dB]
Sensor Noise	1.085	-	-	-
Resting	1.105	0.008	1.019	0.08
Stat. Activity	1.149	0.011	1.059	0.25
Locomotion	1.213	0.129	1.118	0.48

## Data Availability

The majority of data processing was performed in MATLAB. The MATLAB scripts for the proposed methods can be found here: https://github.com/pietertry/TunedBeam-IMU-basedActivityRecognition, accessed on 26 February 2025.

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
