# Peer review of "Vibration-Based Non-Contact Activity Classification for Home Cage Monitoring Using a Tuned-Beam IMU Sensing Device"

_sensors, 2025, doi:10.3390/s25082549_

Round 1
Reviewer 1 Report
Comments and Suggestions for Authors
The authors develops a non-contact method of monitoring an activity of an animal in a home-cage. The method comprises an IMU sensor and an algorithm for classification of the activity type. The system is simpler and cheaper than analogues while achieving comparable performance. The system is verified against ground truth data supplied by videocamera-based solution, which is a traditional approach here. In general the work is interesting and described clearly and many details are presented. However, some important blocks are missing.
1. Decision tree in figure 4 is clear and seems logical. However nothing is said about the way it was obtained. Did the expert build it completely out of imagination? Or group of experts? If so, how can we prove it is correct and adequate to the problem? Or some statistics was involved? Then, what is the statistics and how did the expert process it? Or this was some automated clusterization? Please explain.
2. Figure 5 contains block-scheme of the algorithm. Infrastructure is clear. However the nature of almost all blocks is not revealed. What are the parameters of MLDWT deconstruction? What phase adjustment is, concretely? Please provide formulas. What is Amplitude transformation? Please provide formulas. And so on for each of the block. These are necessary parts of the method and they should be presented in detail.
Reviewer 2 Report
Comments and Suggestions for Authors
The manuscript develops a sensor that could detect mouse activities. A convolutional neural network - long short-term memory method is proposed. The research topic is interesting. The manuscript is organized well. The result is convinced. Thus, the manuscript can be accepted.
Reviewer 3 Report
Comments and Suggestions for Authors
Thanks for the opportunity to read the article.
A few comments on the work.
1. Lines 177 - 180
It is necessary to check the numbering of the figures and links to them. For example, the link to Figure 6 is unclear
2. How were the thresholds (Figure 3) determined for classifying the activity?
3. The authors know that the signal from the sensors will be noisy. Why not use filters to clean the signal from noise, for example, the Kalman filter?
4. It is not very clear why LSTM is used if it is possible to classify pieces of a time series using wavelets and extracting further information from them. How necessary was LSTM here?
5. In addition to the activities defined in the article, there are many other activities. Including those that are very important when assessing the behavior of an animal. The authors did not assume that it would be necessary to add to the monitoring system
Round 2
Reviewer 1 Report
Comments and Suggestions for Authors
The authors have convincing answers and appropriate revisions to the comments. The papers can be published now.